# Post-Failure Dynamics of Rainfall-Induced Landslide in Oltrepò Pavese

**Sauro Manenti [1,2,*]**, **Andrea Amicarelli [3]**, **Nunziarita Palazzolo [1]**, **Massimiliano Bordoni [4]**, **Enrico Creaco [1,2,*]** and **Claudia Meisina [2,4]**

[1] Department of Civil Engineering and Architecture, University of Pavia, via Ferrata 3, 27100 Pavia, Italy; nunziarita.palazzolo01@universitadipavia.it

[2] Interdepartmental Centre for Water Research, University of Pavia, via Ferrata 3, 27100 Pavia, Italy; claudia.meisina@unipv.it

[3] Ricerca sul Sistema Energetico-RSE SpA, Department SFE, via Rubattino, 54, 20134 Milan, Italy; andrea.amicarelli@rse-web.it

[4] Department of Earth and Environmental Sciences, University of Pavia, Via Ferrata 1, 27100 Pavia, Italy; massimiliano.bordoni@unipv.it

* Correspondence: sauro.manenti@unipv.it (S.M.); creaco@unipv.it (E.C.); Tel.: +39-0382-985-321 (S.M.); +39-0382-985-317 (E.C.)

**Abstract:** Prediction of landslide hazard risk at hill slope induced by intense rainfall requires the appropriate modeling of the interactions between soil and weather phenomena, leading to failure as well as a reliable prediction of post-failure dynamics. In the peculiar case of fast shallow landslides behaving like dense granular flows, a suitable modeling approach for large and rapid deformations is necessary to estimate potential related damage. The impact force exerted by the leading edge of the earth-flow on the downstream structure should be estimated for both damage prediction and design of effective protection measures. In this paper, a free open source 3D research code based on standard weakly compressible smoothed particle hydrodynamics (WCSPH) method is validated by modeling a full-scale rainfall-induced shallow landslide which occurred in Oltrepò Pavese (Northern Italy). The code allows resolving the vertical velocity gradients, potentially providing a more reliable representation of the landslide dynamics and impact force. Mechanical parameters are consistent with average soil characteristics, avoiding calibration analysis. The final landslide profile is compared with an experimental survey for model validation, showing good fit. Influence of uncertainties of geotechnical parameters on the landslide front velocity and impact force on the downstream wall is evaluated.

**Keywords:** rainfall-induced shallow landslide; post failure dynamics; soil-water interaction; validation; non-Newtonian fluid; WCSPH; natural hazards

## 1. Introduction

Post-failure shallow landslide dynamics may be characterized by opposite behaviors. In the case where low displacement rates occur at a narrow shear zone just below the ground surface, some simplified mathematical approaches may be adopted in the engineering practice to describe effectively the landslide dynamics [1]. In the case of fast rainfall-induced shallow landslides, the dynamics is more complex to predict if the run-out starts as shallow rotational-translational failure subsequently changing into earth-flows. This kind of landslide is triggered by intense rainfall events inducing water infiltration at slopes that increases the volumetric water content and pore water pressure, thus worsening the slope stability [2]. Therefore, a reliable assessment of landslide susceptibility requires, among the other things, proper definition of the hydro-mechanical features, as well as rainfall characteristics

considering recent climate trends affecting rainfall and intense storm events [3]. A rainfall-induced shallow landslide represents one of the most common natural hazards in some areas of the world [4,5]. Ongoing research is being carried out for defining a new integrated hydrogeological model to assess shallow landslides and flood prone areas at catchment scale in Oltrepò Pavese, in order to predict their spatial and temporal occurrence, and to develop early-warning strategies (ANDROMEDA project, funded by Fondazione Cariplo) [6].

The above described dynamic behavior of fast rainfall-induced shallow landslide is due to the large water content. As a consequence, this kind of landslide behaves like a dense granular flow, and suitable modeling approaches should be adopted for handling large spatial and temporal displacement gradients. Meshless particle methods could be helpful for this purpose. Among the numerous types of meshless particle methods, the smoothed particle hydrodynamics (SPH) method [7] is successfully applied to simulate complex multiphase flows with impact and shock [8,9], involving fluids with high-density ratio [10] as well as non-Newtonian fluids [11–14]. These problems are of great concern in the applied engineering dealing with water related natural hazards [15], such as landslide induced tsunami in artificial reservoir [11] and intense rainfall-induced shallow landslides [16]. The weakly compressible (WCSPH) code SPHERA v.9.0.0 (RSE SpA) [17] is a free open-source research software that has been validated on various application fields including floods with transport of solid bodies and bed-load transport; fast landslides and their interactions with water reservoirs; sediment removal from water bodies [18]. The validation of SPHERA by simulating the post-failure dynamics of rainfall-induced fast shallow landslide will be illustrated in this work.

In the analysis for a reliable estimate of the risk level of rainfall-induced landslide hazard, the susceptibility evaluation represents only one of the relevant issues. In fact, relevant dynamic features of landslide run-out, such as width and length of damage corridor, travel velocity, characteristic depth of both moving mass and deposit, should be properly assessed in order to provide a quantitative estimate of the hazard and select appropriate protective measures for risk mitigation [19]. This target may be achieved through the adoption of reliable predictive models providing quantitative information on the destructive potential of the landslide.

There are several approaches that have been successfully adopted for the post-failure dynamic analysis of fast shallow landslides. In the work of [20], the post-failure flow model simulates the moving landslide as single-phase continuum through FLO-2D commercial software that solves numerically the depth-integrated flow equations with the finite volume approach. The SPH model in [21] allows one to predict the path, velocity, and depth of flow-like landslides following a two-phase approach where the mixture is made up of a solid skeleton with the voids filled by a liquid phase. The mixture dynamics is described by quasi-Lagrangian depth integrated governing equations of mass and momentum balance, and the pore pressure dissipation equation. The model was used to reproduce a catastrophic event occurred on May 1998 in the Campania region (Italy), showing the relevant role of geotechnical parameters (especially the fluid phase and angle of internal friction) for the reliable prediction of the run-out distance, velocity, and height of the landslide. A proper selection of the values assigned to these parameters assured the best agreement with the field observations.

When the initial average depth is comparable with the horizontal length and width of the collapsed soil, as in the present case, significant variations in both thickness and vertical velocity profile may occur along the flow direction, advising against the adoption of depth-integrated models. Furthermore, a numerical model that does not require the calibration of relevant parameters in the landslide post-analysis may be used for run-out prediction in a risk analysis. In this work, the shallow landslide is simulated with a 3D code that allows resolving the velocity gradient in the vertical direction, potentially providing a more reliable representation of the landslide dynamics and impact force. No tuning of the relevant geotechnical parameters has been carried out to reproduce the final landslide profile with suitable accuracy. The impact force on the vertical downstream wall can be calculated for both quantitatively evaluating landslide risk level and supporting the design of protective structures.

The following Section 2 illustrates briefly the relevant features of the numerical model adopted for the post-failure dynamics of the landslide and the case study. Section 3 presents and discusses the simulations results, including: the model validation on a full-scale rainfall-induced shallow landslide; the effects of uncertainty of the relevant geotechnical model parameters (i.e., angle of internal friction $\phi$, effective porosity $e$ and solid phase density $\rho_s$) on the leading front celerity and impact force on the downstream wall. Conclusions are illustrated in Section 4.

## 2. Model Description and Test Case Configuration

SPHERA v.9.0.0 holds several numerical schemes, among which an SPH formulation of mixture model for the analysis of dense granular flows consistent with the kinetic theory of granular flow (KTGF). Those features of this scheme relevant for the present study are summarized in the current section; for a complete description, the reader is referred to [22]. The 3D fluid dynamics of both water and soil mixture is obtained by solving numerically the mass and momentum balance equations that are discretized according to standard WCSPH formulation:

$$\left\langle \frac{d\rho_i}{dt} \right\rangle = -\rho_i \sum_{j=1}^{N} \frac{m_j}{\rho_j}\left(\mathbf{u}_j - \mathbf{u}_i\right) \cdot \nabla W_{ij,h} + B_\rho$$
$$\left\langle \frac{d\mathbf{u}_i}{dt} \right\rangle = -\frac{1}{\rho_i} \sum_{j=1}^{N} \frac{m_j}{\rho_j}\left(p_i + p_j + \frac{\Pi_{ij}}{\rho_i}\right)\nabla W_{ij} + \frac{\nabla \cdot \boldsymbol{\tau}_i}{\rho_i} + \mathbf{g} + \mathbf{B}_u \tag{1}$$

In Equation (1), the symbols assume the following meaning: $\mathbf{u}$ mixture velocity; $\mathbf{g}$ gravity acceleration; $B_\rho$ and $\mathbf{B}_u$ boundary contributions for mass and, respectively, momentum balance equations according to a semi-analytic approach [23]; $p$ mixture pressure; $m$ particle mass; $\rho$ mixture density; $W_{ij}$ compact supported central kernel function of the relative distance between a generic fluid particle, denoted by index $i$, and each one of the $N$ neighboring particles, denoted by index $j$; $\Pi_{ij}$ Monaghan artificial viscosity term [7] used for the fluid phase solely. The divergence of the mixture viscous stress tensor $\boldsymbol{\tau}$ in the momentum balance (for the mixture solely) is computed as a function of the strain rate tensor $\mathbf{D}$ through the apparent viscosity $\mu_{fr}$ depending on the effective stress $\sigma'$, the second invariant of the strain rate tensor $II_D$ and the angle of internal friction of soil $\phi$:

$$\nabla \cdot \boldsymbol{\tau}^s = \nabla \cdot \left(2\mu_{fr}\mathbf{D}\right)$$
$$\mu_{fr} = \frac{\sigma' \sin\phi}{2\sqrt{II_D}} \tag{2}$$

The time integration is carried out through a second-order Leapfrog scheme calculating the velocity of each particle at mid time-step with respect to both position and density [24]. As the numerical scheme for the solution of discretized mass and momentum balance equations is explicit, the following stability condition must be satisfied for the integration time step $dt$ [25]:

$$dt = \min\left\{C_v \frac{2h^2}{\mu/\rho}; \ CFL\frac{2h}{c+|\mathbf{u}|}\right\} \tag{3}$$

In Equation (3), $h$ denotes the smoothing length related to particle size $dx$, $c$ is the sound speed and CFL denotes the Courant–Friedrichs–Lewy number. A severe time step reduction occurs, caused by the viscosity stability condition when shear rate comes close to zero and the apparent viscosity approaches higher values according to pseudo-plastic behavior (second of Equation (2)). To avoid the unbounded growth of both mixture viscosity and computational time at a very low shear rate, the maximum (or threshold) viscosity $\mu_{max}$ is introduced with a physical meaning [22]. When approaching to zero shear rates, those mixture particles with an apparent viscosity higher than the maximum viscosity are considered in the elastic–plastic regime of soil deformation, where the kinetic energy of solid particles is relatively small, and the frictional regime of the packing limit in the KTGF does not apply. In this condition, the maximum viscosity is assigned to those particles that are excluded from the

SPH computation and considered as fixed particles. When these particles fall inside the neighbor list of a nearby moving particle, suitable physical properties are assumed. The value of the maximum viscosity does not require tuning or calibration. For a specific problem, the value assigned to the maximum viscosity is the smallest value that does not influence the numerical results appreciably. Any increase above this value would only affect the computational time through a reduction of the time step (Equation (3)). This procedure is descried in Section 3.1.

By analogy with the experimental behavior of high polymer solutions, the transition from the frictional regime (i.e., solid particle in motion) to the elastic-plastic regime (i.e., solid particle at rest) can be reproduced within a limiting region in the flow curve that is characterized by a constant value of the apparent viscosity referred to as limiting viscosity $\mu_0$. The limiting viscosity acts as a numerical parameter [26]. The value assigned to viscosity $\mu_0$ should be evaluated through an optimization procedure leading to the definition of the lower value of the limiting viscosity to obtain a suitable accuracy in the results. This approach was successfully tested against a 2D laboratory experiment reproducing, along a representative transversal section of the artificial basin, the run-out of the 1968 Vajont landslide on the left-hand slope [26]. By taking into account landslide interaction with the water (both stored in the artificial basin and filling the saturated landslide portion), the scheme of SPHERA for dense granular flows allowed predicting maximum wave run-up with both suitable accuracy and significant reduction in the computational time.

In the present work, the cited scheme is validated on the post-failure analysis of a rainfall-induced shallow landslide. This landslide occurred during an intense rainfall event on April 2009 in a hilly area of the Oltrepò Pavese named Recoaro valley (north-western part of Italy). Even if SPHERA is a 3D code, its simpler and faster 2D execution has been conveniently chosen for this case: as the landslide is relatively narrow in the transversal direction, in the upper and mid parts of the domain, the flow may be assumed as two-dimensional in the longitudinal vertical plane (Figure 1a).

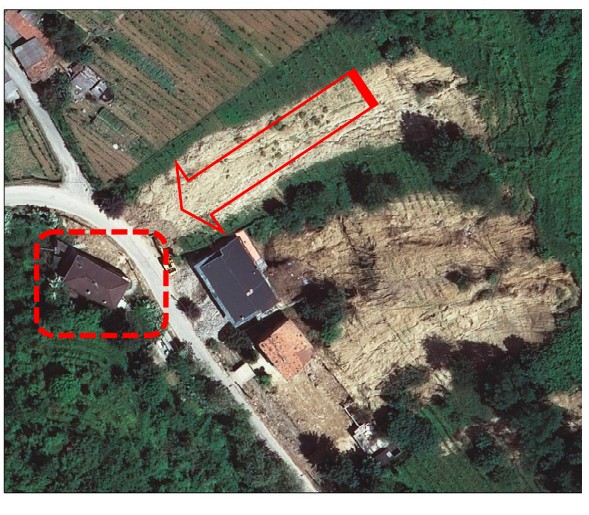　　　　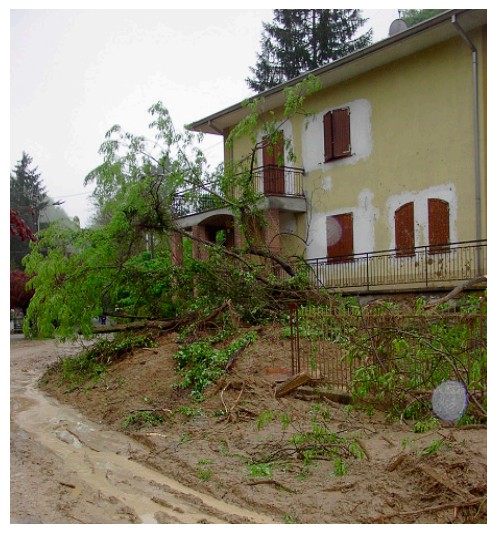

(**a**)　　　　　　　　　　　　　　　　　　　　　(**b**)

**Figure 1.** Rainfall-induced landslide occurred on April 2009 in the Recoaro valley of the Oltrepò Pavese: (**a**) Plan view with the sliding direction (red arrow) and the downstream structure impacted by the landslide (dashed red square), the aerial photograph of the area was taken by Ditta Rossi s.r.l. (Brescia, Italy) on 18 May 2009; (**b**) Detail of the impacted structure showing part of the slide.

After sliding inside an almost straight corridor with a nearly constant width of about 14 m (Figure 1a), the saturated soil impacted against the vertical wall of a building located at the toe of the opposite slope, causing the landslide to stop abruptly (Figure 1b). Some small trees were transported by the earthflow, but they were neglected in the present modeling. The portion of the deposit obstructing

the street on the valley floor was suddenly removed for safety purposes. Anyway, the upstream portion of the deposit was not altered, and survey data of its final profile are available.

The position of the downstream obstacle is not perpendicular to the landslide propagation direction, thus after the impact, the actual landslide behavior near the wall may deviate from the 2D assumption. However, this feature does not affect the landslide characteristics during the run-out phase that are relevant for risk assessment (i.e., front thickness and velocity), but may affect the prediction of impact force that cannot be considered uniformly distributed in the transversal direction. Furthermore, from Figure 1b, the depth of the deposit just in front of the impacted structure can be estimated, and it seems to be almost uniformly distributed in the transversal direction. This information is used as reference for comparison with the landslide simulated final profile. The relevant geotechnical characteristics of the soil involved in the collapse are collected from [5] and summarized in Table 1. These soils are clayey-sandy silts, characterized by close to nil cohesion. The values of the soil features have been used to estimate the input parameters adopted in the validation analysis.

**Table 1.** Summary of the values assigned to relevant geotechnical parameters.

| Relevant Geotechnical Soil Parameters | |
| --- | --- |
| angle of internal friction $\phi$ | 24–26° |
| saturated unit weight $\gamma_{sat}$ | 18–19 kN/m$^3$ |
| saturated volumetric water content $\vartheta_s$ | 0.42–0.46 |

## 3. Results and Discussion

This section illustrates and discusses the results of performed simulations. An overall number of 26 simulations have been carried out to investigate various parameters that are relevant for the modeling, as discussed in the following. Table 2 shows the relevant geotechnical parameters of the rainfall-induced landslide occurred on April 2009 in the Recoaro valley. According to the observations, the soil mass is assumed to be fully saturated at the beginning of the collapse under the gravity force. Furthermore, it is quite reasonable to neglect the soil cohesion as the failure takes place. The speed of sound $c$ in the solid phase is the square root of the ratio between the bulk compressibility modulus and the density. A unique speed of sound is considered, so that each phase respects the weakly compressible approach constraint on the density error [22]. The artificial viscosity coefficient is here slightly smaller than $\alpha_M = 0.10$ [26]: the reference value for the landslides simulated with SPHERA should be considered as $\alpha_M = 0.09 \pm 0.01$.

**Table 2.** Summary of the relevant input parameters.

| Relevant Input Parameters | |
| --- | --- |
| particle resolution $dx$ | 0.25; 0.10 m |
| smoothing length $h$ | 0.325 m; 0.13 m |
| artificial viscosity coefficient $\alpha_M$ | 0.075 |
| sound speed $c$ | 80 m/s |
| solid phase density $\rho_s$ | 2650 kg/m$^3$ |
| angle of internal friction $\phi$ | 24° |
| effective porosity $e$ | 0.42 |
| maximum initial landslide height $h_0$ | 136.7 m a.m.s.l. |
| run-up length on the downstream wall $l_{r\text{-}u}$ | 4.0 m |

First of all, the simulation results involving the effects of both maximum viscosity and limiting viscosity are discussed to explore the trade-off between computational time and accuracy in the results (Section 3.1). Then, the final landslide profile after the stop at the toe of the downstream structure is compared with survey data for model validation (Section 3.2). Finally, the influence of geotechnical

input parameters on the landslide front celerity and impact force, which affect the dangerousness associated with a landslide occurrence, is analyzed (Section 3.3).

### 3.1. Analisis of Viscosity

As explained in the previous section, early simulations have been carried out to define the value of the maximum viscosity $\mu_{max}$ for the problem under consideration. Following the approach tested in previous works [22,26], the initial simulation is performed by assuming a suitably high value that is $\mu_{max} = 1.0 \times 10^6$ kPa. This choice requires the adoption of a suitable initial value of the particle spacing which is conveniently set at $dx = 0.25$ m, based on the problem characteristic length-scale. The effect of limiting viscosity is neglected at this stage of investigation by assigning to it a value greater than maximum viscosity. The resulting rheological behavior is shown in the lower right-hand part of Figure 2: pseudo-plastic behavior is assumed for those shear rates that correspond to an apparent viscosity $\mu_{fr}$ lower than $\mu_{max}$. At each subsequent simulation, the maximum viscosity value is reduced by one order of magnitude down to $\mu_{max} = 1.0 \times 10^3$ kPa. The landslide front celerity is monitored during the run-out until the stop. The main plot in Figure 2 shows the non-dimensional evolution of the landslide front position $\xi$ over non-dimensional time $\tau$, where $x$ is the horizontal abscissa along the sliding direction, $h_0$ denotes the maximum initial landslide height (Table 2), that is, the height of the upper landslide front at $x = 16.20$ m. No noticeable variation in the landslide kinematics can be appreciated, despite the significant reduction in the computational time. An additional simulation is then performed to assess the effect of lowering the maximum viscosity to $\mu_{max} = 0.5 \times 10^3$ kPa. As can be seen from the black dashed curve in Figure 2, this further reduction in the maximum viscosity produces some effects on the kinematics of the landslide front: the final position around time $\tau = 3.5$ is nearly the same as in previous simulations, but the leading front celerity is a little bit smaller in the time interval $\tau = (1.5–3.0)$.

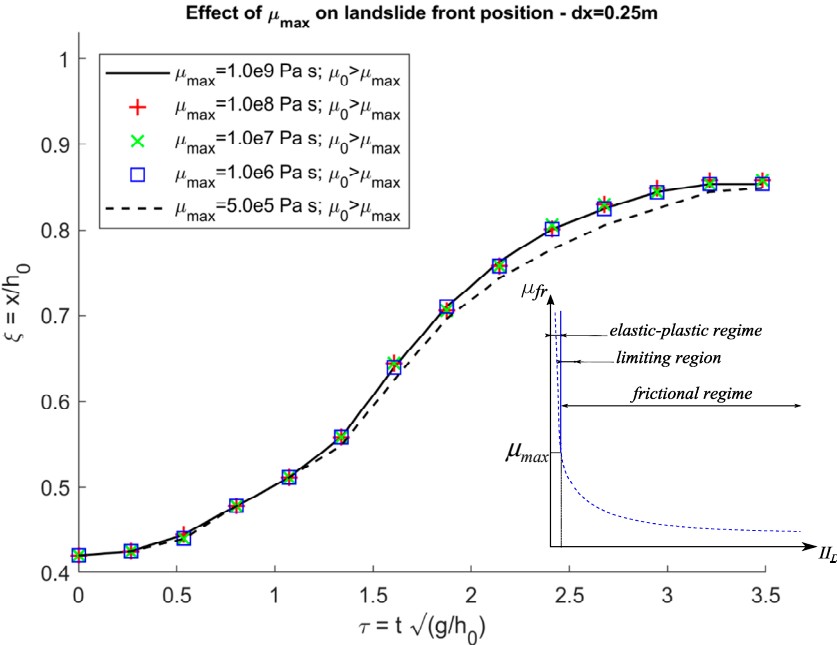

**Figure 2.** Effect of maximum viscosity $\mu_{max}$ on landslide front propagation. Non-dimensional front position $\xi$ as a function of the non-dimensional time $\tau$.

From the simulations performed, it can be concluded that the value $\mu_{max} = 1.0 \times 10^3$ kPa does not affect appreciably the landslide dynamics and allows one to significantly reduce the computational time. Therefore, this value is adopted for the maximum viscosity in subsequent simulations.

As a result of the computational time reduction, the particle resolution was changed from the initial value down to a lower value of $dx = 0.10$ m. A further reduction in $dx$ below such a value does not significantly affect the numerical results while increasing the computational time. Although the value of the maximum viscosity depends on the spatial resolution and the limiting viscosity, the above procedure seems sufficient for the current test case.

The subsequent set of simulations is carried out to define the optimal value for the limiting viscosity, in order to further reduce the computational time while maintaining a suitable level of accuracy of the simulation results. The lower right-hand part of Figure 3 shows the resulting rheological behavior when activating the limiting viscosity $\mu_0$ (red curve). As can be seen, in the limiting region between the transition from the frictional regime to the elastic-plastic regime, the actual value of the mixture viscosity $\mu_{fr}$ is approximated with the constant value $\mu_0 < \mu_{max}$. This approximation allows a further reduction in the computational time through Equation (3), but the amount of approximation that is introduced should be maintained below a suitable threshold.

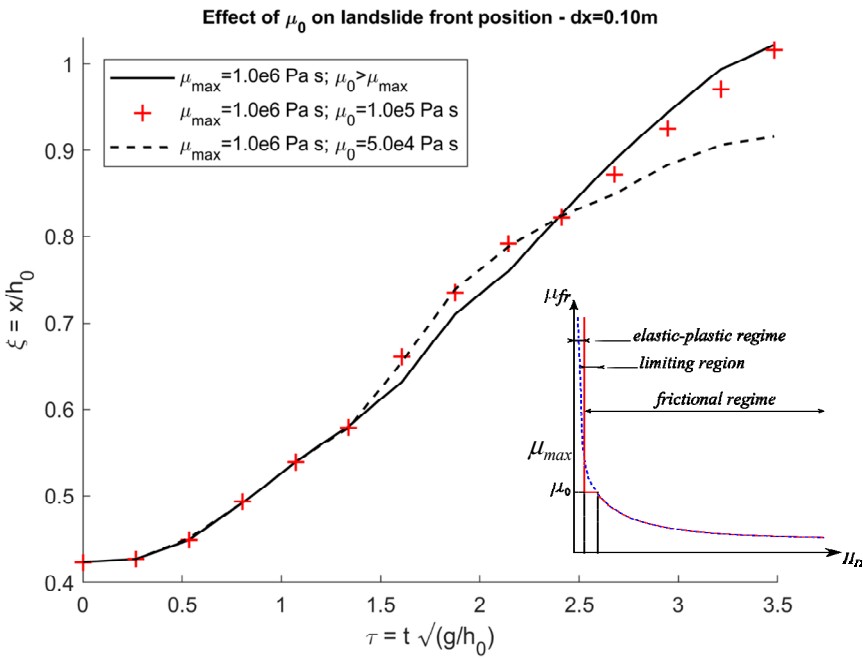

**Figure 3.** Effect of limiting viscosity $\mu_0$ on landslide front propagation. Non-dimensional front position $\xi$ as function of non-dimensional time $\tau$.

The landslide front position versus time is shown in the main plot of Figure 3 when reducing progressively the limiting viscosity below $\mu_{max}$. The vertical downstream wall has been temporarily removed in order not to affect the landslide front kinematics. It can be seen that negligible change in the landslide front celerity compared to the reference curve (continuous black line with $\mu_0 > \mu_{max}$) can be obtained with the assumption $\mu_0 = 0.1 \times 10^3$ kPa (red cross markers). Further reduction in the limiting viscosity to $\mu_0 = 0.05 \times 10^3$ kPa causes an unacceptable reduction in the landslide front celerity.

In conclusion, for the present analysis, the maximum viscosity can be set equal to $\mu_{max} = 1.0 \times 10^3$ kPa without influencing the landslide kinematics, while the limiting viscosity can be conveniently assumed to be equal to $\mu_0 = 0.1 \times 10^3$ kPa. This result represents an optimal compromise between model accuracy and computational time.

### 3.2. Validation

The simulation of the actual landslide occurred on April 2009 in the Recoaro valley after intense rainfall events had been carried out, assuming the geotechnical parameter values in Table 2.

These represent average values for the soil involved in the failure and no calibration was carried out for fitting the numerical results to the observed data.

Figure 4 shows some representative frames of the post-failure dynamics. Upper frames show the initial volume of unstable soil that is obtained by subtracting the pre-event profile to the post-event profile. In the central part of the profile, no significant variation of the topographic surface can be appreciated, probably due to low run-out ratio. Therefore, in this case, soil entrainment was neglected. In the case where soil entrainment represents an important process, it can be simulated with either a scheme for the erosion process that has been implemented in SPHERA [12], or by enlarging the particle numerical domain to the whole granular material potentially subjected to soil entrainment. It can be seen that, in the early phase, at $t$ = 3.0 s, the mass portion close to the landslide front moves faster than the rear portion. Just after the impact against the vertical wall of the downstream building, at $t$ = 13.0 s, the landslide front began decelerating and stopped, while the rear mass portion on the steep slope still maintained a relatively high average speed. In this case, the modeling approach based on the infinite landslide with constant depth and moving at a constant velocity on a constant slope may not be appropriate at all. Instead, the proposed modeling allows for reproducing the velocity gradients along the landslide, potentially providing more reliable representation of the landslide dynamics and the resulting impact force on the rigid wall.

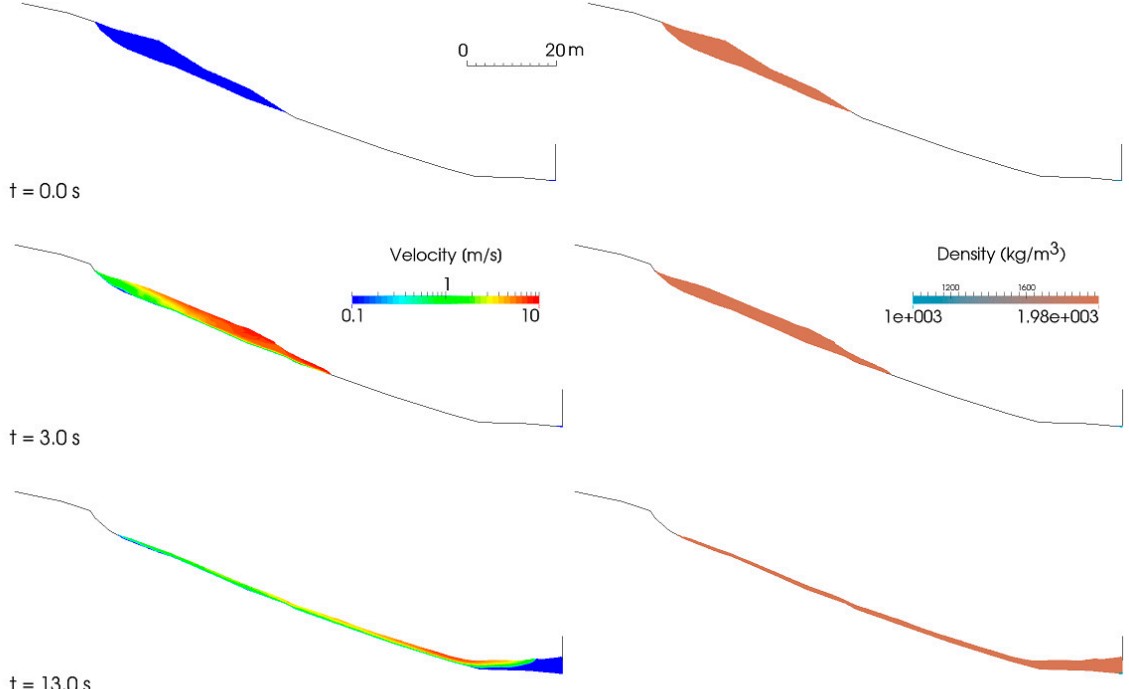

**Figure 4.** Representative frames of landslide run-out and impact. Contour plots show velocity magnitude (left-hand panels) and density field.

The final landslide profile at $t$ = 60 s is shown by the red line in Figure 5. At this stage, the major part of the landslide mass has come to a complete stop. The blue markers represent the experimental points obtained during the post event survey. The comparison shows that numerical results provide a quite acceptable prediction of the final configuration of the landslide. Even if at $t$ = 60 s, the main portion of the landslide body has come to a complete stop, some few particles in the upper part of the sliding profile show a small downstream velocity and probably may compensate at later instants for the slight underestimation of the height of experimental points between abscissa 65 m and 80 m.

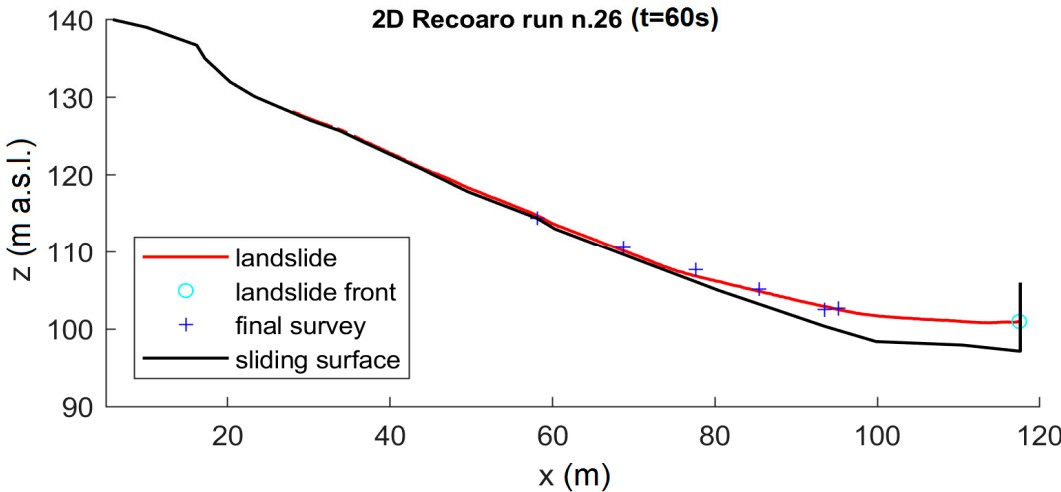

**Figure 5.** Simulated final profile of the landslide at $t = 60$ s (red line). The upper part of the landslide profile is compared with experimental points from the post-event survey (blue markers).

As previously said, the estimate of related risk depends upon several landslide characteristics, such as: width and length of damage corridor, travel velocity, characteristic depth of both moving mass and deposit. In this case study, the width and length of the damage corridor are fixed; the characteristic depth of deposit is suitably well represented; the average travel velocity seems quite acceptable as the run-out duration, from the beginning of the sliding till the impact on the wall, is about 10 s, which is quite reasonable for the considered case. Therefore, it can be concluded that the model allows for obtaining an acceptable estimate of the risk level associated to the landslide.

### 3.3. Influence of Input Parameters

To consider the effects of the uncertainties potentially affecting the relevant geotechnical parameters, their influence on the landslide front celerity and the impact force on the downstream vertical wall has been evaluated from the fluid pressure field over the wall. SPHERA also allows a more accurate assessment of the fluid forces exerted on solid bodies by means of a scheme for the transport of solid bodies [18,24]. The following parameters have been considered in the sensitivity analysis: angle of internal friction $\phi$; effective porosity $e$; and solid phase density $\rho_s$. The investigated range of each parameter has been selected based on the typical range for the soil involved in the simulated landslide, as described in Table 1. In particular, for the angle of internal friction, a variation of $\pm 4°$ is assumed (run 22 and run 23), with respect to the average value $\varphi = 24°$ adopted in the validation analysis carried out with the reference run 26 (Table 2). Concerning the effective porosity, a bigger value of $e = 0.50$ is assumed (run 24). Finally, an increased density value of the solid phase equal to $\rho_s = 2900$ mk/m$^3$ is considered (run 25). The influence produced by the variation of each parameter has been evaluated individually, and the obtained results are discussed below.

Figure 6 shows the evolution of non-dimensional landslide front position $\xi$ versus the non-dimensional time $\tau$. The continuous black line shows the result for the reference run used for model validation. As the wall is assumed to be rigid, the final landslide front position after the wall impact is independent from the values assigned to the model parameters. According to theoretical expectations, a reduction in the angle of internal friction produces an increase in the landslide front celerity (red square marker). Similar behavior is detected in the simulation with increased effective porosity (blue plus marker), where the rise of the water content reduces, as expected, the sliding resistance between mixture particles, resulting in an increase in the landslide speed. In both runs 22 and 24, the impact against the downstream vertical rigid wall is anticipated around time $\tau = 2.3$.

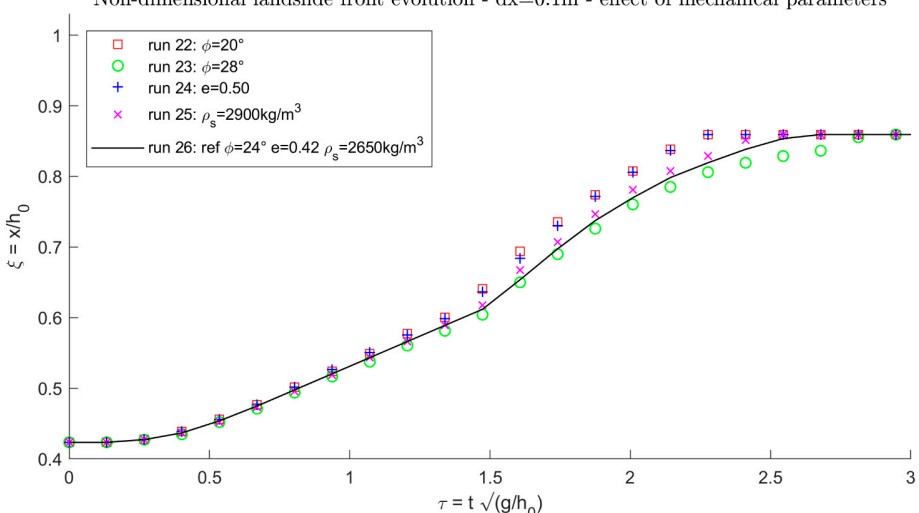

**Figure 6.** Effects of geotechnical parameters variation on landslide front propagation. Non-dimensional front position $\xi$ as function of non-dimensional time $\tau$.

When increasing the solid phase density in run 25, a restrained increase in the landslide front velocity is attained (pink cross markers). The impact with the vertical downstream wall occurs around time $\tau = 2.5$, slightly before the impact time detected in the reference run at about time $\tau = 2.7$.

The increase in the angle of internal friction in run 23 is responsible for an overall reduction in the landslide front celerity, whose impact on the downstream vertical wall occurs at later time close to $\tau = 3.0$. This behavior, which is quite the opposite of the one detected when reducing the angle of internal friction, is in accordance with the formulation in Equation (2), where the frictional resistance between mixture particles increases with $\phi$.

The influence of these parameters on the landslide kinematics affects the dynamic impact as well. The greater the landslide velocity, the greater and earlier the impact force. This is confirmed by the analysis of results in Figure 7, where the non-dimensional resultant force on the wall per unit width versus non-dimensional time is shown for the four simulations in comparison with the reference run. The parameter $l_{r-u}$ represents the run-up length on the downstream wall (Table 2).

The greatest impact force occurs for run 22 with a reduced angle of internal friction; this peak occurs just after the impact instant when the frontal portion of the landslide with the highest kinetic energy has come to an almost complete stop. In the case of run 24, which showed a similar landslide front kinematics to run 22, the peak of force is significantly shifted forward in terms of the impact time and is much lower. This is due to the pronounced fluidic behavior caused by the higher water content: as a result, in run 24, the force acting on the downstream wall is distributed over a wider time span (about $\Delta\tau = 0.8$), resulting in a less steep trend and following a reduced peak.

In both runs 23 and 25, the peak of impact force occurs just after the impact of landslide front on the downstream wall. The magnitude of the peak for run 25 is much greater than the reference run and close to the peak observed in run 24, but the impact force evolves very quickly over time in run 25.

If compared with run 22, it can be seen that the magnitude of peak force is much lower for run 23, due to the highest frictional dissipation that reduces the overall amount of kinetic energy transferred to the wall. This behavior is in accordance with Equation (2).

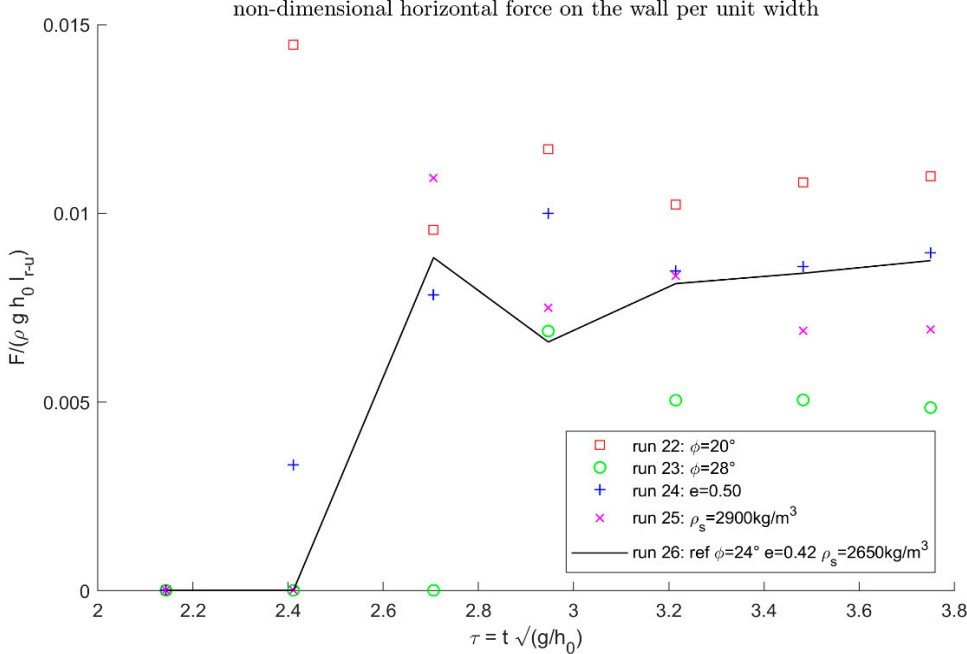

**Figure 7.** Effects of geotechnical parameters variation on the impact force on the vertical downstream wall.

## 4. Conclusions

This work illustrates the validation of the free open source 3D research code SPHERA [17], 2019; [27] based on WCSPH method for modeling a full-scale rainfall-induced fast shallow landslide occurred in 2009 on a hilly area in the North-western side of Italy. Comparisons with on-site experimental data are reported. Owing to the peculiarities of this landslide being characterized by a relatively constant width of the almost straight sliding corridor, the flow may be assumed as two-dimensional in the longitudinal vertical plane, and a simpler and faster 2D execution is conveniently carried out.

In contrast with depth integrated models, the adopted model allows for resolving the vertical velocity gradients of the landslide that are produced by its geometrical characteristics. Furthermore, the resolution of the vertical velocity gradients leads to a reliable representation of the landslide dynamics and, possibly, of the related impact force on the downstream wall.

The characteristic depth of the final deposit is suitably well represented; the average travel velocity seems quite acceptable for the considered case. These results are fundamental for obtaining a reliable estimate of the risk level associated to the landslide.

The time evolution of the resultant impact force on the downstream vertical rigid wall is also computed. This feature may be helpful in supporting the design of reliable protection measures.

No tuning of relevant geotechnical parameters is needed for fitting the numerical results to the observed data from the post-event survey. Input parameters can be obtained straightly from average soil characteristics. Overall good model accuracy is found: this relevant result seems promising for the application of the model to the prediction of fast shallow landslide related risk.

Sensitivity analysis has been finally carried out to assess the possible uncertainties affecting relevant geotechnical parameters. Obtained results are consistent with the theoretical expectations.

In this work, the volume of the soil affected by the landslide can be determined from the comparison between pre- and post-event topography. In order to extend the model applicability to landslide risk prediction, future developments will be devoted to coupling the code SPHERA with a rainfall-induced landslide triggering model, capable of estimating the landslide volume, starting from data concerning rainfall pattern and initial soil conditions.

**Author Contributions:** Conceptualization, S.M.; writing—original draft preparation, S.M.; computer simulations and analysis, S.M.; review and editing, S.M., A.A., E.C., N.P., M.B. and C.M.; funding acquisition, C.M. All authors have read and agreed to the published version of the manuscript.

**Funding:** This work has been in the frame of the ANDROMEDA project, which has been supported by Fondazione Cariplo, grant n. 2017-0677. The support contribution of the RSE author (management of the code SPHERA) has been financed by the Research Fund for the Italian Electrical System (for "Ricerca di Sistema -RdS-"), in compliance with the Decree of Minister of Economic Development 16 April 2018.

**Acknowledgments:** We acknowledge the CINECA award under the ISCRA initiative, for the availability of High Performance Computing resources and support". In fact, SPHERA simulations of the present study have also been financed by means of the following instrumental funding HPC projects: HPCNHLW2, HPCNHLW1. SPHERA v.9.0.0 (RSE SpA) is realised thanks to the funding "Fondo di Ricerca per il Sistema Elettrico" within the frame of a Program Agreement between RSE SpA and the Italian Ministry of Economic Development (Ministero dello Sviluppo Economico). The contribution of the RSE author is made for hire. We also acknowledge: the Ditta Rossi s.r.l. (Brescia, Italy) for the acquisition of the aerial photographs of the study area; Giuseppe Barbero for post event survey data of the landslide profile; Gabriella Petaccia, member of the project ANDROMEDA and leader of WP3 "Development of the integrated hydrological-hydraulic model".

**Conflicts of Interest:** The authors declare no conflict of interest. The funders had no role in the design of the study; in the collection, analyses, or interpretation of data; in the writing of the manuscript, or in the decision to publish the results.

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
