# Peer review of "Post-Failure Dynamics of Rainfall-Induced Landslide in Oltrepò Pavese"

_water, doi:10.3390/w12092555_

Round 1

Reviewer 1 Report

The authors aim to validate the 3D SPH code SPHERA for modelling rainfall‑induced shallow landslides. The proposed approach has potential in improving risk assessment of rainfall-induced shallow landslides in combination with a landslide triggering model. Several comments and suggestions are prepared for the authors below:

  • Page 4, line 150: It would have been better to implement an example with 3D geometry to support validation of the 3D SPHERA code for rainfall-induced shallow landslides. The validity of the 2D assumption in the Oltrepò Pavese case study should be further examined. I agree with the authors that the 2D assumption is representative in the upper part of the domain. However, this assumption should be also examined in the lower part as the position of the obstacle is not perpendicular to the flow, with the final deposition profile not necessarily being uniform in the transversal direction. Please justify this assumption and explain the potential effects on the results. I would encourage authors to consider evaluating the code in future studies on an actual 3D problem with more complex geometry, mass entrainment, and longer runout ratios.
  • Page 4, line 158: "After sliding inside an almost straight corridor with nearly constant width of 1m, the saturated soil impacted against the vertical wall of a building located at the toe of the opposite slope causing the landslide to stop abruptly (Fig. 1b)." This is not clear from Figure 1 as the unstable zone seems to be significantly wider. Please explain this or add additional information.
  • Page 5, table 2: Speed of sound in what?
  • Page 6, line 201: Non-dimensional evolution of the front position is defined relative to x, which is not defined at this point in the paper. Please specify it when introducing the non-dimensional front position.
  • Page 6, line 201: The second parameter of the non-dimensional front position is h0 and it is defined as the maximum initial landslide height. The location of this point is not clear. It could be potentially more useful to refer to the position of the centre of mass of the unstable soil.
  • Page 8, Figure 4: How was the initial volume selected? Additionally, why is there no soil between the initially unstable soil volume and the road?
  • Page 8, Figure 4: Soil entrainment is an important process in modelling rainfall-induced landslides. This case study does not seem to implement this. Is the code capable of modelling this and how would it be implemented?
  • Page 9, Figure 6: All the simulations with the wall (Figures 2 and 6) result with almost the same final values of the non-dimensional front position. What are the effects of the initial volume on the final values of the non-dimensional front position?

Author Response

Please, see attached file "Responses_to_Reviewer_1"

Reviewer 2 Report

This paper pretty well described their research to be published. However, it is hard to find the originality of this paper because the authors only used a commercial code and performed some parametric studies. To increase the impact of this paper, new findings from this study must be presented in detail.

Author Response

See attached file "Responses_to_Reviewer_2"
